# EEGFormer: Towards Transferable and Explainable Large-Scale EEG Foundation Model

**Yuqi Chen[1]\*, Kan Ren[2], Kaitao Song[1], Yansen Wang[1],**
**Yifan Wang[2], Dongsheng Li[1], Lili Qiu[1]**

[1] Microsoft Research    [2] ShanghaiTech University
yansenwang@microsoft.com    renkan@shanghaitech.edu.cn

## Abstract

Self-supervised learning has emerged as a highly effective approach in the fields of natural language processing and computer vision. It is also applicable to brain signals such as electroencephalography (EEG) data, given the abundance of available unlabeled data that exist in a wide spectrum of real-world medical applications ranging from seizure detection to wave analysis. The existing works leveraging self-supervised learning on EEG modeling mainly focus on pretraining upon each individual dataset corresponding to a single downstream task, which cannot leverage the power of abundant data, and they may derive sub-optimal solutions with a lack of generalization. Moreover, these methods rely on end-to-end model learning which is not easy for humans to understand. In this paper, we present a novel EEG foundation model, namely EEGFORMER, pretrained on large-scale compound EEG data. The pretrained model cannot only learn universal representations on EEG signals with adaptable performance on various downstream tasks but also provide explainable outcomes of the useful patterns within the data. To validate the effectiveness of our model, we extensively evaluate it on various downstream tasks and assess the performance under different transfer settings. Furthermore, we demonstrate how the learned model exhibits transferable anomaly detection performance and provides valuable explainability of the acquired patterns via self-supervised learning.

## Introduction

Scalp electroencephalography (EEG) is physiological signal data that provides valuable insight into the human brain activities and has extensive applications in healthcare, e.g., disease diagnosis and medical monitoring (Lawhern et al. 2018; Tang et al. 2021, 2023; Li et al. 2023). Despite the ease of collecting EEG signals, comprehending and interpreting them often requires extensive expertise from medical professionals. To address this challenge, recent research has focused on leveraging self-supervised learning techniques to learn meaningful representations from EEG data (Yi et al. 2023; Wang et al. 2023; Li et al. 2022). These learned representations can then be fine-tuned for various downstream tasks, including seizure detection (Tang et al. 2021, 2023),

abnormal detection (Darvishi-Bayazi et al. 2023), emotion recognition (Yi et al. 2023; Ye, Chen, and Zhang 2022; Song et al. 2021; Li, Wang, and Lu 2021), etc. However, these existing works focus on pretraining upon each individual dataset corresponding to a single downstream task and fail to leverage the power of abundant data. In this paper, our primary interest lies in exploring the potential of self-supervised learning using abundant large-scale unlabeled data without human annotations.

Moreover, explainability is a crucial concern when applying machine learning models to real-world applications (Peng et al. 2022; Ali et al. 2022; Leung et al. 2022), particularly in the healthcare community (Mendoza-Cardenas, Meek, and Brockmeier 2023; Gulamali et al. 2023). Prior research (Tang et al. 2021; Wang et al. 2023) has predominantly relied on end-to-end model learning, which poses challenges for human comprehension. Models that lack explainability have the potential to yield unsafe and irrational outcomes, thereby increasing the risk of severe medical malpractice.

To address the above issues, we introduce EEGFORMER as a solution for large-scale EEG pretraining. Our primary objective is to investigate a discrete representation learning approach (Van Den Oord, Vinyals et al. 2017; Fortuin et al. 2018; Peng et al. 2022; Esser, Rombach, and Ommer 2021) specifically designed for EEG pretraining. We provide evidence that the utilization of vector-quantized Transformer (Vaswani et al. 2017) model can learn universal representations on EEG signals with adaptable performance on various downstream tasks compared to the conventional mask reconstruction strategy (Nie et al. 2022). Furthermore, the learned codebook and the discrete indices provide explainable outcomes of the useful patterns within the data.

The contribution of the paper can be summarized as below:

- We propose a novel pretraining strategy for EEG data. EEGFORMER adopts a discrete representation learning algorithm along with reconstruction loss.

- We harness the plentiful EEG data available in the TUH Corpus (Harati et al. 2014) to construct a foundational EEG model. This marks the pioneering effort in pretraining with a massive 1.7TB EEG dataset.

- We conduct a comprehensive analysis of the pretrained foundation model EEGFORMER, evaluating its performance on four downstream corpora sourced from the

---
\*The work was conducted during Yuqi Chen's internship at Microsoft Research. Correspondence to Kan Ren.

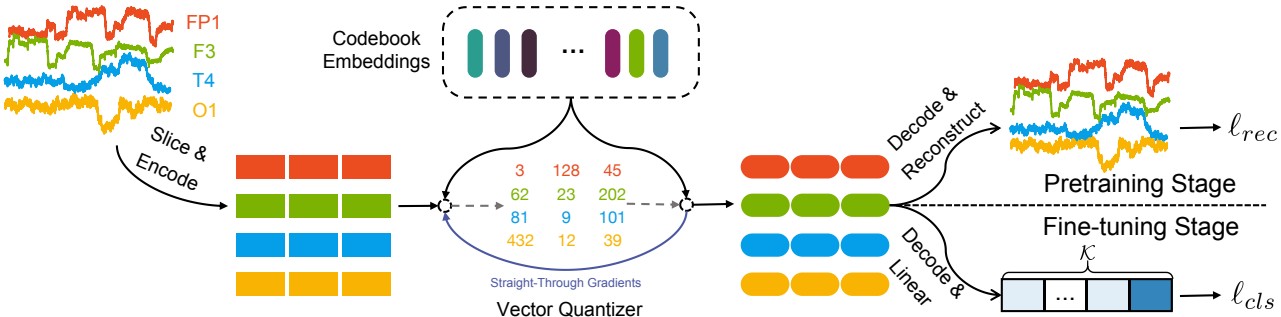

Figure 1: Overview of EEGFORMER. Initially, multi-variate EEG signals are segmented into patches, which are then passed through a Transformer encoder. Subsequently, a vector-quantized model is employed to generate discrete indices. These indices are then fed into a shallow Transformer decoder.

TUH corpus. Additionally, we explore its transferability by applying it to the Neonate dataset (Stevenson et al. 2019) for neonatal seizure detection.

- We provide an in-depth analysis of the learned codebook and demonstrate that the pretraining algorithm can provide transferable and explainable representations.

## Related Work

**Pretraining for Time-Series Data**    Self-supervised learning for time-series data is a highly significant research hotspot. Many non-Transformer models have been developed to learn the representation of time series (Franceschi, Dieuleveut, and Jaggi 2019; Tonekaboni, Eytan, and Goldenberg 2021; Yue et al. 2022; Eldele et al. 2021). Recently, (Nie et al. 2022) introduced a Transformer-based approach that segments time series into patches, which leads to promising outcomes across various forecasting datasets. Furthermore, researchers are growing interested in utilizing pretrained large language models (LLMs) to enhance time series analysis (Zhou et al. 2023; Gruver et al. 2023). These methods are mainly on forecasting tasks and lack practical considerations of the model adaptation to different downstream tasks.

**Pretraining for EEG data**    Electroencephalograms (EEGs) are widely employed for diagnosing neurological, and psychiatric, as well as in brain-machine interface applications. In the field of EEG signals, self-supervised learning has emerged as a promising approach (Tang et al. 2021; Jiang et al. 2021; Kostas, Aroca-Ouellette, and Rudzicz 2021). SeqCLR (Mohsenvand, Izadi, and Maes 2020) introduces a set of data augmentations for EEG and extends the SimCLR (Chen et al. 2020) framework to extract channel-wise features on time-series EEG data. MMM (Yi et al. 2023) focuses on spatial and topological modeling of EEG data and breaks the boundaries between different EEG topologies. However, these methods rely on end-to-end model learning, which lacks explainability. In this paper, we propose a new pretraining strategy that can provide an explainable representation. Moreover, these methods either apply self-supervision within the same dataset or test for a single downstream task, which cannot fully unleash the power of the self-supervised pretraining paradigm. In this paper, our approach diverges

the existing methods by leveraging the extensive multiple datasets of different tasks for pretraining purposes.

## EEGFORMER: Vector-Quantized Pretraining Transformer for EEG Data

This work aims to present a novel pretraining algorithm to derive a universal, transferable, and explainable EEG foundation model. In this paper, we focus on learning temporal patterns among multi-channel EEG data. Specifically, we view EEG data as a multi-variate time series data, i.e., $X \in \mathbb{R}^{L \times C}$, where $L$ represents the length of the time series, and $C$ represents the number of channels (or variates) [1]. Our primary goal is to develop a self-supervised learning algorithm that optimally leverages unlabelled data while enhancing explainability. To accomplish this, we introduce a customized vector-quantized pretraining approach designed for EEG data, as illustrated in Figure 1. EEG signals can be encoded into discrete tokens, enabling explanation through the analysis of these tokens, as is discussed in experiments. During the fine-tuning stage, the model and the codebook can be further fine-tuned to integrate specific domain-specific knowledge. In the subsequent subsections, we will provide a detailed description of the overall framework, including the preprocessing, EEG slicing, encoding module, decoding module, training algorithm, and fine-tuning processes.

**Feature Preprocessing**    Converting EEG signals to the frequency domain is a common preprocessing technique. Inspired by (Tang et al. 2021), given a time domain EEG signals, we perform fast Fourier transformation (FFT) to obtain frequency domain amplitude as input features.

**Slice & Encode**    To pretrain a time-series tokenizer, we first apply instance normalization to the frequency domain inputs. Then, we split each univariate time series into non-overlapped (or overlapped) segments (Nie et al. 2022). Specifically, for each variate (or channel), i.e., $x_c \in \mathbb{R}^L$ for the $c^{\text{th}}$ variate. Denote the patch length as $P$ and the stride as $S$, the patching

---
[1]We mitigate the sample rate discrepancy by resampling the EEG data to a uniform rate of 250 Hz. Further, our analysis focuses on fixed-length 12-second EEG data following (Tang et al. 2021). Thus, throughout the experiment, $L$ equals to 3000.

process will generate a sequence of patches $\mathbf{x}_c \in \mathbb{R}^{P \times N}$, where $N = \left(\lfloor \frac{L-P}{S} \rfloor + 2\right)$ indicates the number of patches. Given the input EEG data $x_c \in \mathbb{R}^{P \times N}$ for $c \in [1, \ldots, C]$, it is necessary to add position embedding before input to the Transformer encoder. Specifically, we map the dimension to $D$ via learnable weight matrix $\mathbf{w}_p \in \mathbb{R}^{P \times D}$ and adopt learnable position embedding, i.e., $\mathbf{w}_{pos} \in \mathbb{R}^{N \times D}$. Hence, the input vector is given by $\hat{x}_c = x_c^\top \mathbf{w}_p + \mathbf{w}_{pos}$. Finally, we forward $\hat{x}_c$ into a stack of Transformer encoder layers in a channel-independent manner (Nie et al. 2022).

**Vector Quantizer** The vector quantizer looks up the nearest neighbor in the codebook for each patch representation $\boldsymbol{h}_i$. Let $\{\boldsymbol{v}_1, \boldsymbol{v}_2, \ldots, \boldsymbol{v}_K\}$ denote the embeddings in the codebook. For the $i^{\text{th}}$ patch, its quantized code is calculated as $\boldsymbol{z}_i = \arg\min_j \|\boldsymbol{h}_i - \boldsymbol{v}_j\|_2$, where $j \in \{1, 2, \ldots, K\}$. After quantizing the hidden vectors to discrete tokens, we obtain the codebook embeddings $\boldsymbol{V}_z = \{\boldsymbol{v}_{z_i}\}_{i=1}^N$.

**Pretraining Stage: Decode & Reconstruct** We further forward the codebook embeddings from the vector quantizer into a shallow Transformer model (Peng et al. 2022). Upon passing through the decoder model, each variate generates an output denoted as $\hat{h}_c \in \mathbb{R}^{N \times D}$. We map the outputs to the same shape as the input through $\mathbf{w}_o \in \mathbb{R}^{D \times P}$ and $\mathbf{b}_o \in \mathbb{R}^P$, i.e., $x_o = \hat{h}_c \mathbf{w}_o + \mathbf{b}_o$. Finally, we reshape the output to match the shape of $X$, denoted as $X_{rec}$. The pertaining objective of EEGFORMER for each sample $X \in \mathcal{D}$ is to minimize

$$\ell_{rec} = \|X_{rec} - X\|_2^2 + \|\operatorname{sg}[\boldsymbol{H}] - \boldsymbol{V_Z}\|_2^2 + \|\boldsymbol{H} - \operatorname{sg}[\boldsymbol{V_Z}]\|_2^2 ,\tag{1}$$

where $\operatorname{sg}[\cdot]$ stands for the stop-gradient operator which is an identity at the forward pass while having zero gradients during the backward pass (Van Den Oord, Vinyals et al. 2017) [2].

**Fine-tuning Stage: Decode & Linear** To facilitate downstream fine-tuning, we utilize the pretrained model weights of both the encoder and the decoder modules. After obtaining the outputs $\hat{H} \in \mathbb{R}^{C \times N \times D}$ from the decoder model, we concatenate all the outputs and transform them into $c \in \mathcal{R}^{\mathcal{K}}$, where $\mathcal{K}$ denotes the number of classes for the classification task. The loss function for the fine-tuning stage is:

$$\ell_{cls} = -\log c_l + \|\operatorname{sg}[\boldsymbol{H}] - \boldsymbol{V_Z}\|_2^2 + \|\boldsymbol{H} - \operatorname{sg}[\boldsymbol{V_Z}]\|_2^2 ,\tag{2}$$

where $l$ is the label of the sample.

## Experimental Results

**Datasets Description** We pretrain our model on the Temple University EEG Corpus (TUH Corpus) [3], which has collected over 1.7TB of unlabelled EEG data that are suitable for pretraining. We evaluate our model on five downstream datasets. i) TUAB corpus for abnormal detection of EEG data. ii) TUAR corpus for classifying artifacts. iii) TUSL corpus for classifying slowing events. v) TUSZ corpus for

---

[2]In Eq. (1), $\boldsymbol{H}$ denotes the hidden vectors for all the variates, whereas $\boldsymbol{h}$ stands for a single variate. Similarly for $\boldsymbol{Z}$ and $\boldsymbol{z}$.

[3]https://isip.piconepress.com/projects/tuh_eeg/

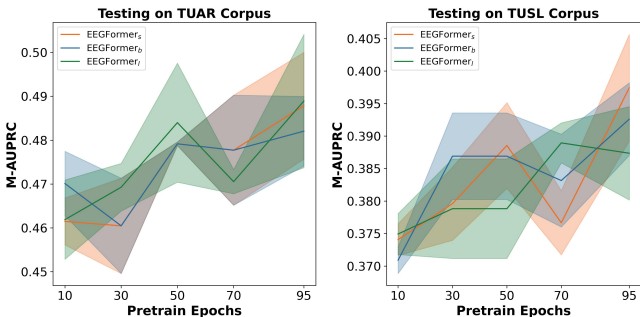

Figure 2: Influence of pretrain epochs on two TUH corpus.

seizure detection. vi) Neonate dataset (Stevenson et al. 2019) for neonatal seizures detection. Notably, the Neonate dataset is not a subset of the TUH dataset. Therefore, we consider the transferability of our pretraining strategy.

**Parameter Setting** We vary the encoder layers from 6 to 12, and the codebook size, i.e., $K$, from 512 to 2048. The decoder is a 3-layer Transformer. We set $D$ to 128. Specifically, EEGFORMER $_s$ adopts a 6-layer encoder and $K = 512$, EEGFORMER $_b$ adopts an 8-layer encoder and $K = 1024$, and EEGFORMER $_l$ adopts a 12-layer encoder and $K = 2048$.

**Compared Baselines** We compare EEGFORMER with several baselines specifically for EEG data. i) EEGNet (Lawhern et al. 2018) adopts a fully convolution network for EEG data. ii) TCN (Bai, Kolter, and Koltun 2018) adopts a dilated convolutional neural network. iii) EEG-GNN (Tang et al. 2021) adopts a graph neural network for capturing spatiotemporal dependencies in EEGs. v) GraphS4mer (Tang et al. 2023) further adopts structured state space models or multivariate biosignals. Additionally, we also compare EEG-FORMER with self-supervised baselines. BrainBERT (Wang et al. 2023) adopts neural signal processing techniques for producing superresolution time-frequency representations and pretrain with mask reconstruction loss.

**Evaluation Metrics** For detection tasks, we adopt the area under the receiver operating characteristic (AUROC) and the area under the precision-recall curve (AUPRC) for evaluation. For multi-classification tasks, we adopt macro AUROC (M-AUROC) and macro AUPRC (M-AUPRC) for evaluation.

**Main Results** The experimental results presented in Table 1 clearly illustrate the effectiveness of our pretraining strategy in both in-dataset and transfer settings. Quantitatively, compared with the best baseline results, EEGFORMER $_l$ achieves a **9.02%** improvement on the Neonate dataset and a **13.23%** on the TUSZ under the AUPRC metric. Additionally, we conduct experiments with different model sizes. Specifically, EEGFORMER $_s$ and EEGFORMER $_b$ demonstrate an average AUROC of 0.822 and 0.829, respectively, as well as an average AUPRC of 0.575 and 0.574, respectively.

**Influence of Pretrain Epochs** We conducted experiments to examine the impact of pretraining epochs on various downstream corpora. The results of these experiments are illus-

Table 1: Experimental results on various downstream tasks. Within the table, * indicates a multi-classification task.

| Model | Pretrain | Metric | TUAB | TUAR* | TUSL* | TUSZ | Neonate |
|---|---|---|---|---|---|---|---|
| EEGNet | ✗ | (M-)AUROC | 0.841 ± .011 | 0.752 ± .006 | 0.635 ± .015 | 0.820 ± .030 | 0.793 ± .019 |
| | | (M-)AUPRC | 0.832 ± .011 | 0.433 ± .025 | 0.351 ± .006 | 0.470 ± .017 | 0.499 ± .044 |
| TCN | ✗ | (M-)AUROC | 0.841 ± .004 | 0.687 ± .011 | 0.545 ± .009 | 0.817 ± .004 | 0.731 ± .020 |
| | | (M-)AUPRC | 0.831 ± .002 | 0.408 ± .009 | 0.344 ± .001 | 0.383 ± .010 | 0.398 ± .025 |
| EEG-GNN | ✗ | (M-)AUROC | 0.840 ± .005 | 0.837 ± .022 | **0.721 ± .009** | 0.780 ± .006 | 0.760 ± .010 |
| | | (M-)AUPRC | 0.832 ± .004 | **0.488 ± .015** | 0.381 ± .004 | 0.388 ± .023 | 0.419 ± .021 |
| GraphS4mer | ✗ | (M-)AUROC | 0.864 ± .006 | 0.833 ± .006 | 0.632 ± .017 | 0.822 ± .034 | 0.719 ± .007 |
| | | (M-)AUPRC | 0.862 ± .008 | 0.461 ± .024 | 0.359 ± .001 | 0.491 ± .001 | 0.374 ± .013 |
| BrainBERT | ✓ | (M-)AUROC | 0.853 ± .002 | 0.753 ± .012 | 0.588 ± .013 | 0.814 ± .009 | 0.734 ± .019 |
| | | (M-)AUPRC | 0.846 ± .003 | 0.350 ± .014 | 0.352 ± .003 | 0.386 ± .018 | 0.398 ± .027 |
| EEGFORMER $_l$ | ✓ | (M-)AUROC | **0.876 ± .003** | **0.852 ± .004** | 0.679 ± .013 | **0.883 ± .005** | **0.833 ± .017** |
| | | (M-)AUPRC | **0.872 ± .001** | 0.483 ± .014 | **0.389 ± .003** | **0.556 ± .008** | **0.544 ± .026** |
| Improvement | | (M-)AUROC | +1.39% | +1.79% | -6.18% | +7.42% | +5.04% |
| | | (M-)AUPRC | +1.16% | -1.03% | +2.10% | +13.23% | +9.02% |

trated in Figure 2, Specifically, the results indicate that a longer pretraining period leads to notable enhancements in the performance of the downstream tasks.

**Compared with Other Settings** Table 2 compares the performance of EEGFORMER $_l$ using fine-tuning, linear probing, and supervising from scratch. By just fine-tuning the model's prediction head, i.e., linear probing), the performance of our model is already comparable with the supervised model, i.e., GraphS4mer. Specifically, EEGFORMER $_l$ with linear probe outperforms GraphS4mer by **1.73%** on the TUAR dataset under the M-AUPRC metric. Thus, we demonstrate that serves as a strong foundation model for EEG data. Furthermore, fine-tuning consistently surpasses the performance of both supervised learning and linear probing, demonstrating the effectiveness of large-scale pretraining.

Table 2: Linear probe results on TUSL and TUAR corpus. Within the table, Sup stands for supervised learning from scratch, FT stands for self-supervised and fine-tuned, and LP stands for self-supervised and linear probing.

| Model | Type | Metric | TUAR | TUSL |
|---|---|---|---|---|
| GraphS4mer | Sup | M-AUROC | 0.833 ± .006 | 0.632 ± .017 |
| | | M-AUPRC | 0.461 ± .024 | 0.359 ± .001 |
| EEGFORMER $_l$ | Sup | M-AUROC | 0.822 ± .012 | 0.703 ± .033 |
| | | M-AUPRC | 0.447 ± .015 | 0.374 ± .003 |
| EEGFORMER $_l$ | LP | M-AUROC | 0.827 ± .000 | 0.657 ± .017 |
| | | M-AUPRC | 0.469 ± .002 | 0.359 ± .003 |
| EEGFORMER $_l$ | FT | M-AUROC | 0.852 ± .004 | 0.679 ± .013 |
| | | M-AUPRC | 0.483 ± .014 | 0.389 ± .003 |

**Towards Seizure Localization** After the pertaining state, each EEG signal is discretized into multiple indices denoted as $I \in [1, \ldots, K]^{C \times N}$. To perform seizure detection in the TUSZ corpus using these pretrained indices, we first extract n-gram features for each data (e.g., 2-gram, 3-gram, and 4-gram). Next, we adopt a naive Bayes classifier based on n-gram features. Notably, we achieve an AUPRC of 0.292

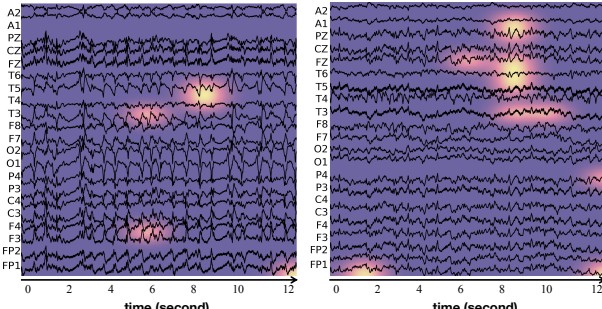

Figure 3: Explanation results from naive Bayes model.

and an AUROC of 0.741, without the need for fine-tuning the pretrained weight. Additionally, we extract the top-3 significant features with high posterior probability leading to seizure events, from the naive Bayes model. Figure 3 presents two cases, where the highlighted regions indicate the localization of seizures. It is worth noting that in the right figure, the highlighted segments correspond to the spike and slow wave complex in all the frontal lobe (Fz), parietal lobe (Pz), and temporal lobe (T3, T6), which indicates an epileptiform discharge (EPSP) followed by the refractory period of the affected neuron population after the large and synchronized neuron EPSP, which is often treated as one of the most important patterns for the diagnosis of epilepsy and the onset of a seizure event. Hence, these patterns are significant in enhancing the explainability of the pretrained model.

## Conclusion

In this paper, we present a novel method called EEGFORMER for self-supervised learning using large-scale EEG data. Our approach learns a discrete codebook and representations of EEG signals simultaneously. We extensively evaluate our pretraining algorithm on various downstream tasks to demonstrate its effectiveness. Additionally, we conduct an analysis to highlight the explainability of our pretraining model.

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
