# OpenReview forum: "EEGFormer: Towards Transferable and Interpretable Large-Scale EEG Foundation Model"
_AAAI.org/2024/Spring_Symposium_Series/Clinical_FMs — AAAI 2024 SSS on Clinical FMs_

### Official Review · Reviewer_LxqK · 2024-02-22
**Interesting work on EEG foundation model**

**Rating:** 9
**Confidence:** 4

**Review:**

This work introduces a self-supervised learning model for EEG data analysis aimed at improving transferability and interpretability. While the approach is innovative, several critical aspects require scrutiny:

1. The concept of leveraging self-supervised learning for EEG data is not novel. The paper must delineate clearly how EEGFormer diverges from and improves upon existing models like BrainBERT or SeqCLR in terms of architecture, learning efficiency, or application to diverse tasks.

2. While the paper asserts that EEGFormer offers interpretable outcomes (which I think it is explainability rather than interpretability), it also falls short of providing a comprehensive framework or quantitative measures for interpretability. The paper should incorporate case studies or comparisons with expert analyses to substantiate these claims.

3. The work needs to show a few-shot learning performance in order to be a foundation model.

---

### Official Review · Reviewer_zQs8 · 2024-02-22

**Rating:** 7
**Confidence:** 3

**Review:**

## Summary
In this work, the authors present EEGFORMER, a foundational model for electroencephalography (EEG) data. They present a new pretraining method for EEG data, that works as follows: first, EEG signals are segmented into patches and passed into a Transformer encoder. Then, they apply a vector-quantized model to convert the patch representations into discrete indices, which as subsequently fed into a Transformer decoder, with the objective to reconstruct the input. The authors apply EEGFORMER in 5 downstream tasks, showing good performance and transfer. Moreover, they show that the representations learned can also be highly interpretable.

## Strengths
- The self-supervised approach the authors propose is promising, and has the potential to efficiently utilize the vast amounts of raw EEG data available.
- The idea to encode the EEG signals into quantized vectors can push the model towards learning interpretable representations, e.g. similar signal patches may be mapped into the same quantized encoding, that may then help us interpret predictions, as the authors show in an experiment.
- The empirical evaluations demonstrate good performance on all downstream tasks tested, and seem transferable.

## Weaknesses
- It would be good to further explore the transferability and interpretability of EEGFORMER's representations, by performing further experiments on additional cases and corpora, to verify if the authors' observations truly generalize.

## Overall Assessment
The paper proposes a novel foundational model and pretraining method for EEG data, and shows strong downstream results and promising interpretability of the representations. It has the potential to pave the way for utilizing large amounts of unlabeled EEG data for various downstream tasks.

---

### Official Review · Reviewer_YWnc · 2024-02-23
**A well written paper that perform their task of making a foundation model capable of solving multiple tasks well and mostly better that state of the art.**

**Rating:** 7
**Confidence:** 4

**Review:**

Clarity:
The authors present a well written paper stating the current litterature and why their work is an advancement in the field of EEG pretrained models stating that their model achieves better performance and is able to generalize across different tasks via finetuning.
Originality: The work combine many state of the art methods in the space of foundation models and apply them to a large scale foundation model trained on EEG data. A left out dataset reserved for validation that has no connnection with the TUH training dataset.
Significance: The performance shows a clear advancement within this space compared to previous supervised and self-supervised models.
Quality: the authors sometimes miss explanations on datasplitting and other common practices that need to be present to ensure. The fundamentals should be included in the script.

Major points
1.	Figure 1 should have a better explanation emphasising subfigures a and b
2.	the finetuning paradigm is not clearly defined. You should consider explaining the way you constrained updating the weights during end-to-end fine-tuning if you did so to avoid misconceptions.
3.	In table 2 you present the AUROC and AUPRC of only a subset of the datasets you have avaliable and you show that with the model architecture you have created there is little difference between traning self-supervised and trainning supervised. I would want you to show the performance on the Neonate dataset to demonstrate the difference between supervised or self-supervised traning.
4.	There is very little disscussion of the results and interpretation of for example figure 3 showing the intrepetation of the naïve Bayes model. Here you seem to highlight the areas for seazures but you do not specify any output from the model.
5.	You need to specify the datasets used for pre-training, fine-tuning, and testing to ensure that there is no lekage from training to testing
Minor points
1.	In table 1, there is little reason to present all the EEGFormer variants as they have very similar performance. I would suggest that you proceed with only the EEGFormerl  for simplicity and just state the other tests in the text.
Pros
•	The authors find high effectiveness in their model training a codebook to represent EEG with subsequent finetuning to solve interesting tasks such as: abnormal EEG detection, classifying EEG artifacts, classifying EEG slowing events, seizure detection, and neonatal seizures detection.
cons
•	Authors do not show the perfomance of finetuning styles for models on all datasets
•	Auphors do not provide the way they split the training, test, and validation data giving no indication that the
•	The authors presend very little discussion on their results, thus making the intrepetation of their results hard to understand out of the gate.
•	The embeddings seem to be highly dependent on large amounts of fine-tuning in order to perform well.